# The Role of Coffee Organizations as Agents of Rural Governance: Evidence from Western Honduras

**Oscar Meza Palma** [1,2,]*[iD], **José M. Díaz-Puente** [1][iD] and **José L. Yagüe** [1][iD]

1   Universidad Politécnica de Madrid, Escuela de Ingeniería Agroalimentaria y Biosistemas (ETSIAAB),
    Avda. Puerta de Hierro 2, 28040 Madrid, Spain; jm.diazpuente@upm.es (J.M.D.-P.);
    joseluis.yague@upm.es (J.L.Y.)
2   Universidad Nacional Autónoma de Honduras, UNAH-CURC, Boulevard Suyapa,
    Tegucigalpa 11101, Honduras
*   Correspondence: oscar.meza@unah.edu.hn; Tel.: +34-504-9949-3438

**Abstract:** Territorial governance is the development strategy that encourages the integration of different actors in the rural environment around common interests. Producer organizations emerge as the appropriate means to unify leadership and consensus to overcome market barriers. These producer networks also influence other dimensions of development, to give way to true governance processes. This paper examines the notion of governance agents associated with the production fabric and the extent to which these agents impact the production efficiency and the level of well-being of those who inhabit the territory. For this purpose, the determinants of a governance model are analyzed in a coffee-growing territory made up of 92 municipalities, located in western Honduras. The analysis is based on a panel of data on the number of cooperatives, rural boards, production parameters, and endowments of public goods grouped into 16 indicators at the municipal level. The data set was subjected to structural equation modeling (SEM), given its statistical capacity to explain complex interrelated phenomena. The main result is the definition of a governance model associated with the coffee territory. This model describes an endogenous pattern of interactions between its four components. This statistical configuration broadens the understanding of the role that governance agents play in fostering a kind of virtuous circle in favor of rural development.

**Keywords:** territorial governance; governance's agents; rural cooperatives; coffee production; structural equation modeling (SEM)

## 1. Introduction

In the rural context of developing countries, agriculture continues to be the backbone of the economy, about 2.6 billion people directly rely on agriculture for livelihoods [1]. But in the current neoliberal economic model, there are market barriers associated with price and quota instability and imbalances in bargaining power, which prevent agriculture from contributing enough to overcome the levels of poverty that characterize these populations [2,3]. Based on the theories of endogenous development and social capital, it has been argued that qualities associated with territorial identity, shared landscape, and leadership are intangible assets that could be mobilized to stimulate associativity and mutual collaboration around common interests [4,5]. The aspirations for quality of life and the predominant sources of wealth in each territory are the attractions that could motivate the organization of people. These interaction networks are the muscle of territorial governance (TG) [6,7].

Recent studies recognize the importance of rural cooperatives and producer associations as sources of social capital, in terms of cohesion, knowledge exchange, and mutual collaboration networks [8,9]. These rural alliances between producers have a significant influence on many associated farmers,

who can thus benefit from reduced transaction costs, achieve greater bargaining power, and better access to financial resources; also through training to achieve a greater administrative capacity of their farms [10–13]. Cooperative links are not limited to the associated group because, in their management of new markets and benefits, they build alliances with both the private sector (wholesalers and intermediaries) and government entities [14,15]. In addition, because they have solidarity community purposes, these organizations go beyond the merely productive and commercial to assume objectives of the common welfare of their territories of influence, something that leads them to join social networks that acquire shared visions on issues of community development [16–18].

These networks of actors that interact with each other and with the government around projects of collective benefit are what different authors call TG [19,20]. It contains the idea of governance because the consensus and goals achieved as a result of the coordination of multi-sectorial efforts remain in the political arena [21,22]. Then, it is territorial because most of the actors who manage to join the networks have a sense of belonging to the territories where they live or work [23,24].

In rural areas, cooperatives and associations of local producers are among the traditional institutions with great influence to exercise leadership and intermediation with and through local governments [25,26]. The success of these organizations associated with the predominant production fabric in each territory encourages the formation of other networks of community actors that seek access to better welfare conditions [27,28]. For this reason, this network of actors is defined as the agent of governance. Since, in this way, resources or knowledge can be shared through daily interaction and trust mechanisms that lead community groups to cooperate with each other, beyond formal procedures or frameworks [14].

The success of governance reported in various territorial contexts has led to the realization of different studies that seek to characterize this social phenomenon [29]. Traditionally, these studies have focused on measuring the contributions and impact left by governance processes. These evaluations have been based on performance or progress indicators of certain factors associated with the social or economic well-being of a specific place or territory. More recently, some authors have recognized the systemic nature of governance, in terms of the different economic, social, political, and cultural dimensions of the territory that are influenced during and through the governance construction processes [30,31]. This new line of research recognizes TG as a complex phenomenon where different variables and factors interact with each other and affect each other in dependency and interdependence relationships [32].

In this sense, this study assumes an analysis of the systemic nature of governance that occurs in the context of a coffee-growing region in western Honduras. More specifically, the empirical proposal focuses on studying variables of the functioning of cooperatives and associations of coffee producers that operate in the study territory. Within the conceptual framework, these organizations act as agents of a kind of coffee governance or are territorially associated with coffee production. The questions to answer are: How is it that these organizations manage to build this network and give rise to coffee governance? With what other territorial dimensions do they link? What implications would this governance have for long-term rural development? These questions are approached empirically but also reinforced with theoretical evidence in order to provide the best possible explanation of the systemic character of governance described above.

This analysis modality tries to go beyond the mere evaluation of territorial governance solely for the directly or indirectly measurable benefits, but also advance in the understanding of how governance unfolds in terms of space and sustainability, as well as the role that governance agents play as triggers of the process. The empirical work carried out has allowed the identification of four latent variables whose pattern of systematic functioning could explain a specific governance model associated with a coffee-growing territory.

The results suggest that this analysis methodology can be replicated not only in coffee-growing contexts but also to other territories with economies based on their own production fabrics.

## 2. Theoretical Background

Approaches to TG and its emergence in rural areas are provided. This discussion opens the way to study hypotheses.

### 2.1. Agents of Rural Governance

In 2015, about 783 million people in the world were living below USD 1.90 a day [1]. It is argued that the decreasing capacity of governments to reduce poverty has been the condition that leads rural populations to become increasingly involved in the search for their own solution strategies [9,22,32]. Therefore, it is postulated that in the governance modality, the leadership of the communities would imply the promotion of practices of association, cooperation, and co-government in networks with those private and social organizations. In these alliances, the government is still a necessary actor but its managerial role is no longer the epicenter of social leadership [23,33,34].

This explains the emergence of local initiatives of different kinds around interests of great significance to communities, such as access to water, health, improvement of economic means of subsistence, or even the environmental problems [27,35–37]. This wave of empowerment allows people to articulate development initiatives in a more informed and inclusive way. Such an approach is contrary to the traditional top-down style as it is guided by the government [24,26,38]. Morgan argues that the capacities of those who inhabit the territory are "a necessary condition so that the decision-making power that exists in politics is translated into the power to transform" [39]. While Pike highlights that "the resources and assets rooted in the territory are likely to integrate more into the local production fabric, inciting less dependence on exogenous economic interests" [40].

This coincides with reports of production improvements in economic activities like agriculture, handicrafts, and forestry that are associated with cooperatives, local government, rural savings banks, and other groups. These groups are understood as the agents (trainers) of a governance network [41–43]. In this sense, it can be determined that there are governance practices that originate from the private, the public, and the social sector [5,28,44]. Some authors think that in rural areas the networks that emerge from the production sector seem to be more stable, but that other networks of actors will necessarily be integrated as expansion occurs [8,29,45].

This is the case of territorial governance associated with the coffee sector of the primary farm link of the production chain, developed by tens of thousands of producers who make up the specificity of a rural coffee community [46–48]. Their organizational expressions arise on a territorial base of coffee-growing municipalities attracted by the common interest of the affiliated producers to face the problems of price instability, the opportunistic behavior of the traders, also for the exchange of knowledge on farm management [17,18,49]. The integration of other territorial actors, such as local governments and community leaders, would be motivated by agendas, interests, and conflicts that coincide in the territory they share. In the development of these governance networks, positive externalities are generated from efforts and unified resources that end up having an impact on the direct and indirect benefit of the population [38,50,51]. How these associations produce and articulate these governance networks is relevant from the endogenous development perspective to design increasingly efficient policies and management tools [11,52,53].

### 2.2. The Systemic Character of Governance

The following describes how coffee activity influences the other dimensions related to territorial development and how these dimensions interact to form a territorial governance model that generates externalities in favor of local development.

All the dynamics of sociocultural relations in the coffee-growing territory revolve around the coffee economy. Therefore, the cultivation of coffee, more than being a business, represents the lifestyle and defines a characteristic identity of the population [10,48,54]. In this way, the integration of different actors occurs because both production and community life occur inextricably [30,55].

The construction of coffee governance from the production base emerges as an endogenous transformative strategy to create systemic relationships between territorial actors [12,16,48]. Cooperatives and associations arise to develop: (1) organizational and coordination capacities to stay informed and jointly manage technical and logistical problems [26,56]; (2) commercial capacities necessary to avoid unfair transactions and access new markets, even more specialized [29,57]; and (3) social capacities necessary to demand improvements in the well-being in their territories [11,31,58,59].

This conceptual perspective plus the expert consultations in the coffee territory under study, allow us to configure what the pattern of a governance model of the study area can be.

In this framework (Figure 1), the governance agents (GA) and the idea of production efficiency (PE) are manifested in those coffee-growing municipalities where producers are more motivated by better farm yields and income, [55,60,61]. During the process, they acquire greater soft skills of managerial and cooperative leadership, increasing their bargaining power and access to new markets with a range of actors at the local, national, and international scale [10].

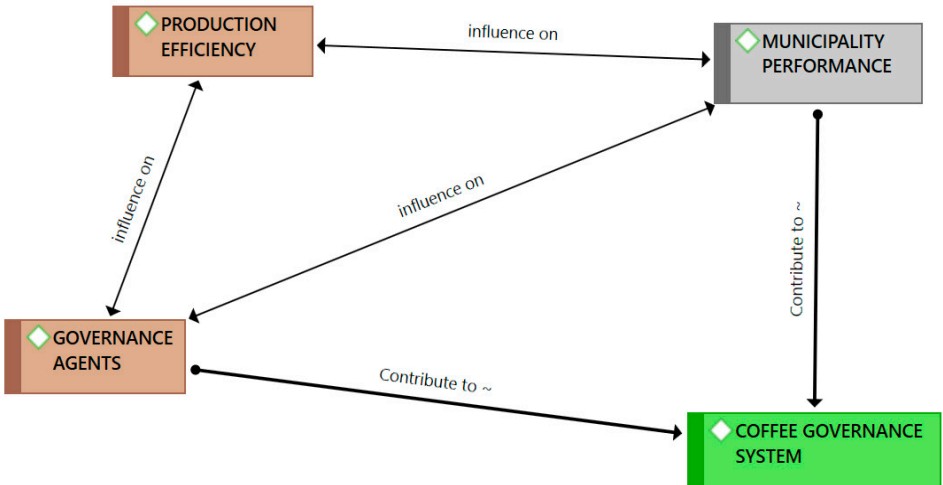

**Figure 1.** TG model associated with coffee production. The author.

Coffee organizations tend to take other steps necessary for the general welfare of producers, such as electrification, drinking water networks, and opening of roads, schools, and health centers. Therefore, they extend their objectives and actions towards the well-being of the families of their affiliates and the conglomerate of farm wage earners, in this way urging and energizing the role of local governments and other development entities that feel more pressured to respond. As a result, there is access to greater PE, coverage of public goods, which is manifested in improvements in the public services provided by the municipality [50,62]. Those measurements that can be captured from these organizations, for example, participation, association, and cooperative indicators are proposed as GA. The effects of these organizations are measured by indicators such as improvements in export capacities, volumes, and harvest quality; and these indicators are used under the concept of PE (Table 1).

**Table 1.** Constructs and indicators identified from the theoretical model under study.

| Constructs | Indicators | Initials | Operative Concept |
|---|---|---|---|
| Governance Agents | Cooperativism | Coop | Index based on the number of existing cooperatives in the municipality |
| | Associativity | Asc | Index based on the number of Producers/Cooperative members |
| | Organization | Org | Index based on the number of the existing coffee boards in the municipality |
| | Coffee participation | Part | Index based on the number of producers members of coffee boards |
| | Municipal public investment | Inv P | Percentage dedicated to public investment from the municipal budget |
| | Municipal financial autonomy | Autn | Municipal solvency to invest with own resources |
| Production efficiency | Productive vocation | VProd | Index based on the number of producers by the municipality |
| | Rural municipal population | Prm | Percentage of rural population based on total population |
| | Contribution in municipal production | AProd | Index based on registered harvest volume by the municipality |
| | Productivity index | Pdvd | Index based on production volume per planted area. Coffee quintal/block |
| | Exportable index | Exp | Percentage of production that can be exported |
| Municipality performance | Drinking water coverage index | Agu | Percentage coverage of drinking water networks in the municipality |
| | Electric coverage index | Elct | Percentage coverage of electrical networks in the municipality |
| | Road investment index or rural roads | InvV | Monetary volume of investment roads by the municipality |
| | Housing deficit | DefV | Homelessness or deficit index in rural space |
| | Health-education endowment | Dot | Proportion of schools and health centers by the municipality |

Note: Once the indicators are entered in the AMOS software, its interface gives them the name of the variables, so they will be used interchangeably later.

TG has also been represented indirectly through indicators of social well-being, shaped by the amount of public infrastructure that each coffee municipality has managed to acquire over time (municipality performance). That is why it is argued that an improvement in governance is seen as a real contribution to development [40,60,63].

Based on this approach, two hypotheses are formulated in this research:

**Hypotheses 1 (H1).** *The role of GA in certain coffee-growing territories favors PE.*

**Hypotheses 2 (H2).** *The performance of the municipality in the acquisition of public goods for its population is influenced by the action of GA who seek productive returns in coffee-growing territories.*

It is thought that where the governance footprint already exists, other associative forms are motivated to emerge, interrelate and unify purposes that define a virtuous circle that looks towards comprehensive and long-term development [23,46]. These qualities would open the way to a governance system associated with the production fabric of each territory [64].

## 3. Materials and Methods

### 3.1. Study Setting

Honduras is the fourth largest coffee exporting country in the world [65]. Exports exceeded 1.5 million dollars in the 2017–2018 harvest. Coffee contributes 8.5% to national GDP and 20% of total agricultural exports [66]. It carries out this production activity in 220 delimited municipalities in six coffee regions of the country. Some 125,000 rural families participate directly, a figure that increases to 1.2 million if an average of five members per family is considered, plus wage earners [61].

The coffee business is a family heritage asset. Of these farms, 92% are considered small farms that do not exceed 5 ha per family [46,65,66]. This is very different from other export crops, where all or most of the value chain belongs to transnational companies. However, it is well known that the primary production of unprocessed coffee fruit is what these rural producers mainly sell, this being a disadvantage because much of the added value remains in the subsequent processing and export links of the chain. A lucrative business that historically is under the control of large national merchants. Precisely the choice of the coffee region of western Honduras as the study area (Figure 2), was because it is where there is more a movement of small producers to access international markets more directly. In addition, this region, made up of 92 municipalities, registers a greater production intensity and contributes 48% of the country's total production where cooperatives, savings banks, and rural boards have positioned themselves mostly as associative practices [47,49,67].

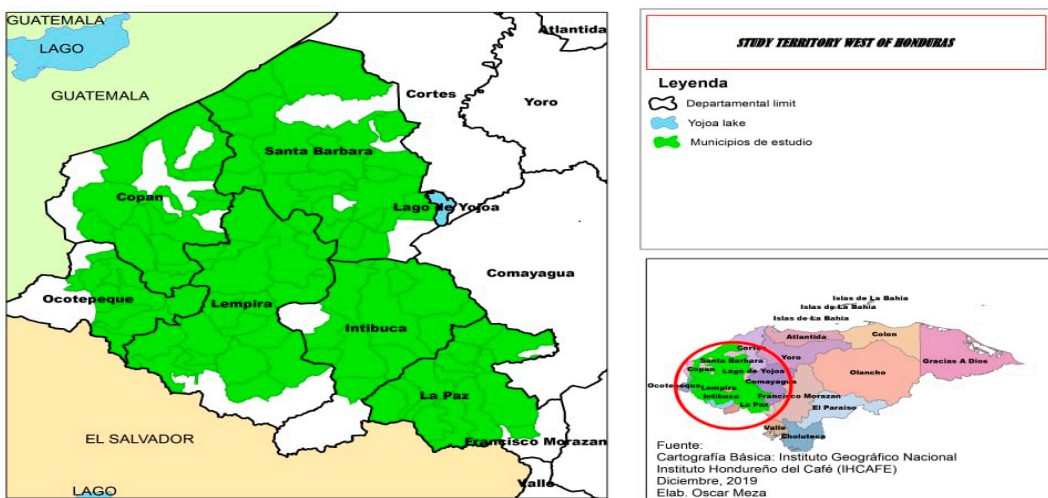

**Figure 2.** Study area, mapped by the author using Arc Gis/2018.

### 3.2. Methodological Strategy

A combination of two complementary research instruments was implemented to test the hypotheses and deepen the explanation of results. The first is eminently quantitative, defined by the SEM model, derived from multivariate statistics. SEM examines dependency and codependency relationships between directly observed variables (panel data). As a result of this algebraic interaction, those latent variables are revealed (called TG constructs, here hypothesized) that otherwise could not be measured, because they cannot be observed directly [45,68].

The qualitative part was conducted through the ATLAS.ti Software (version 8.0). The qualitative is intended to place quantitative results on the ground, something that has been achieved thanks to semi-structured surveys applied to GA, in which different actors from the study area participate. Their points of view and experiences regarding the producer organizations that are part of it were collected to integrate into the analysis of the governance system under study.

The SEM analysis was carried out using software called Analysis Moments of Structures (AMOS, version 24.0). AMOS has the attributes required in the theory of structural equations. It runs a range of

tests with which it is possible to test the model's hypotheses [69,70]. Among the tests executed are the maximum likelihood test of the Chi-Squared (CMIN), the comparative adjustment index (CFI), the measure of the degree of variance and covariance that is explained by the model (GFI), and the adjustment measure that compares the mean of differences in the degrees of freedom that can occur in the population.

A panel of 16 indicators (observable variables) were subjected to SEM analysis. These data capture different dimensions of the performance of the municipalities and are theoretically associated with four unobservable variables also called constructs (see Figure 1): municipal performance, governance agents, production efficiency, and coffee governance system. In the first phase, data were collected from the 92 municipalities in the study area. It consists of disaggregated secondary sources of information from public and private sector institutions: the National Statistics Institute [71], Honduran Coffee Institute [66], and national coffee producers associations. Each construct contains indicators that could best explain it regarding its function within the analyzed governance model. This arrangement along with its operational concepts is shown in Table 1.

### 3.3. Data processing and Exploratory Factor Analysis (EFA)

The data set was first ordered in the program SPSS, version 24.1, giving way to the first test of the measurable quality of the variables, prescribed in statistical theory as a prerequisite to the analysis of the variance normality ANOVA [72]. This data must meet centrality assumptions such as the normality of variance through the Kolmogorov–Smirnov Test KMO, homoscedasticity (Levene's test), and multi-collinearity or degree of correlation between variables [73]. Additionally, SEM analyses also comply with a protocol called Exploratory Factor Analysis (AFE) necessary to simplify the data set.

In EFA, the measurement of latent variables is defined based on the observed variables. In other words, the main components defined by the variance error (Kaiser test) and the factor loads or degree of correlation (Pearson test) that save the set of variables are extracted [69]. The three components calculated in EFA that appear in Table 2 (of results) were labeled with the concept that best corresponds to the constructs (concepts of latent variables) to be analyzed. Since the function of EFA is to reduce the set of variables to those components that have similar characteristics in the range of variance. That is components that behave homogeneously [69].

**Table 2.** Factorial loads by method of extraction of main components.

| Variables | Components | | | Variables | Components | | |
|---|---|---|---|---|---|---|---|
| | 1 | 2 | 3 | | 1 | 2 | 3 |
| Productivity (Pdvd) | 0.907 | - | - | Productive vocation (Vprod) | - | 0.888 | - |
| Cooperativism (coop) | 0.812 | 0.389 | - | Organization (Org) | 0.318 | 0.734 | |
| Municipal Autonomy (Autn) | 0.799 | - | - | | | | |
| Associativity (Asc) | 0.790 | 0.436 | - | Contribution in municipal Production (AProd) | 0.487 | 0.711 | |
| Municipal Public Investment (inv p) | 0.678 | - | - | Housing déficit (DefV) | - | - | −0.856 |
| Roads Investment (InvV) | 0.645 | 0.487 | - | Electricity coverage (Elct) | - | - | 0.810 |
| Export (Exp) | 0.635 | 0.383 | 0.335 | Water network coverage (Agu) | 0.308 | - | 0.729 |
| Coffee Participation (part) | - | 0.905 | - | Health-Education Endowment (Dot) | - | - | 0.703 |

Rotation methods Varimax with Kaiser normalization. The rotation has converged in five iterations.

### 3.4. Confirmatory Factor Analysis (CFA)

This stage is run in the AMOS software. To run the confirmatory factor analysis CFA, first, enter the data matrix filtered by the centrality tests [72]. The data entry to the software is guided by the analysis of trails or the "path diagram" that is proposed considering the components extracted in the EFA analysis (Table 2), also based on the theoretical design of the coffee governance system proposed in this paper (Figure 1). This allows us to corroborate the quality of measurements made in EFA, leading to a greater contrast of the specified hypotheses [68]. The interface provided by AMOS simultaneously performs two analyses: the measurement determined for CFA and the measurement that established the correlation between the variables and the factors or constructs. The closer they get to one, the closer the association between variables [73].

AMOS also analyzes the structure of the "path diagram" to be represented spatially. This structure could then undergo some position adjustments that try to correspond to the best model option that meets the reliability and precision tests [45]. In the "path diagram", it can be seen that the rectangles represent measurable variables and the ovals represent the components (Figure 3).

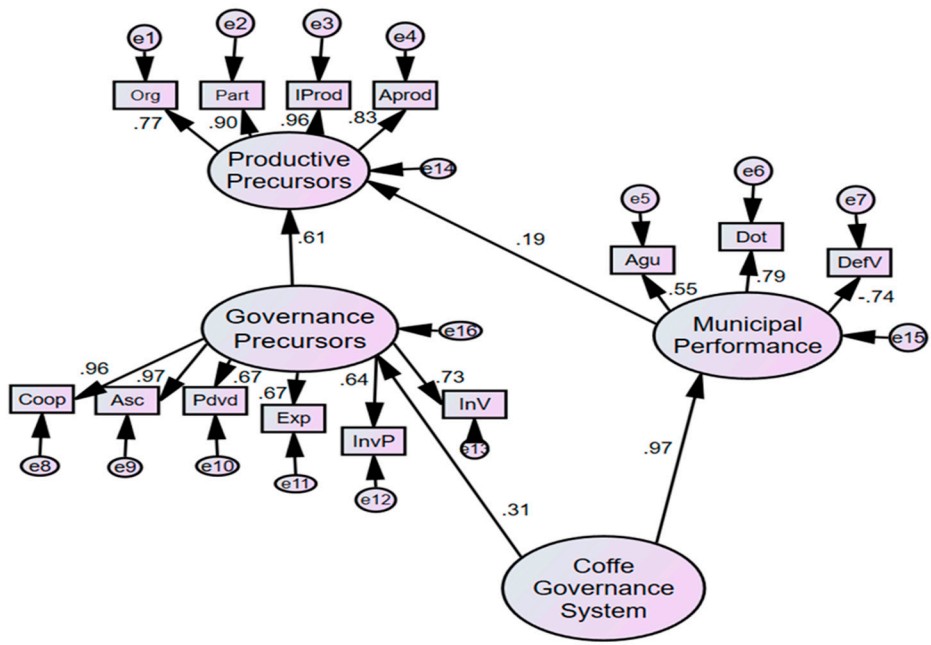

**Figure 3.** SEM Analysis Coffee Governance Model.

### 3.5. Qualitative Analysis

This qualitative part, in addition to explaining the information obtained in the SEM analyses, was useful to better understand how governance agents build their networks of trust, around which values they define their negotiations and consensus. The qualitative analysis attempts to capture a dialogic relationship based on the exchange of knowledge and aspirations [74]. In this way, the statistical analysis takes advantage of the heterogeneity in the responses of the interviewed actors to achieve a greater understanding of their results. Since the qualitative paradigm is not about accepting a single truth. Rather, it is about including the different but shared thinking of those who inhabit a social and geographical space [75]. This part was carried out under the criteria of representativeness and ethical restrictions to ensure the confidentiality of the person interviewed. In this way, 76 interviews were applied throughout 10 representative municipalities in terms of the production and large presence of coffee organizations. Those coffee producers who are affiliated with the coffee organizations scattered in the municipalities of the study area were interviewed, among them managers of cooperatives and coffee boards. Also, professionals associated with the agronomic and administrative management of coffee cultivation that operates in the territory were included in this sample. This work was carried out

between the months of October to December 2019. All interviews were recorded on paper, captured in Spanish, and translated into English by the authors.

Semi-structured interviews were used focused on the same four constructs that were subjected to analysis in the quantitative part, trying to collect the perceptions and positions of the interviewee regarding these issues. Thus, the richness of the daily experience of the interviewees can be better captured and triangulated with the quantitative part, which improves the validity and reliability of the study findings [76,77]. The flexible modality of the interview allowed us to deepen and detail, in the opinion of the producers, those key points that determine the TG. In the interviews, convergence was detected on issues such as the importance of being rooted in the territory, what things motivate associating with others, the intrinsic value of being a coffee grower, and trust in the institution. All the information collected was organized into seven categories fed by a selective codification of the different concepts that revolve around the idea of governance since the grounded theory recognizes as theoretical foundations the collective and similar thinking of those who share the territory [78]. This part was guided by ATLAS.ti software version 8.0 (see Figures 4 and 5), which organizes and synthesizes the information collected through semantic networks (diagrams) that highlight the living expression of the informant, providing a very useful didactic illustration to contrast with the quantitative part [74].

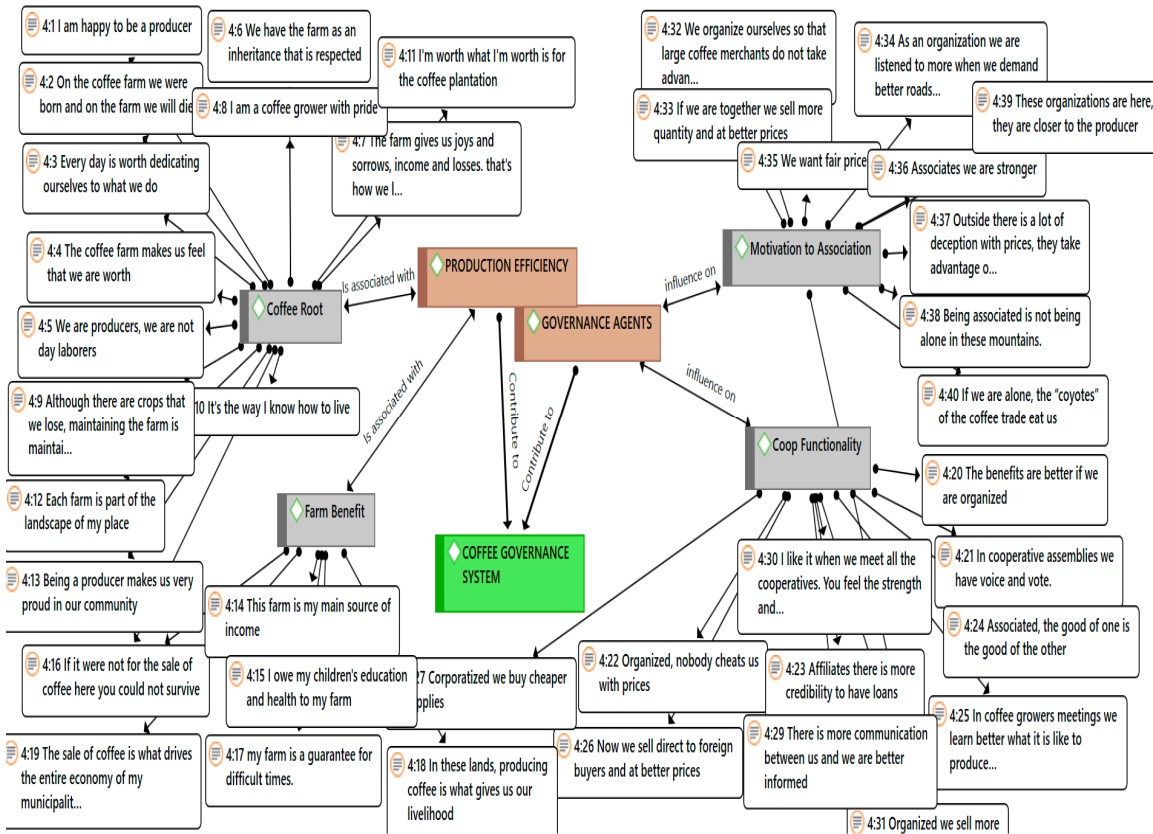

**Figure 4.** Elements of governance analyzed, translated into English by the author from interviews processed in ATLAS.ti.

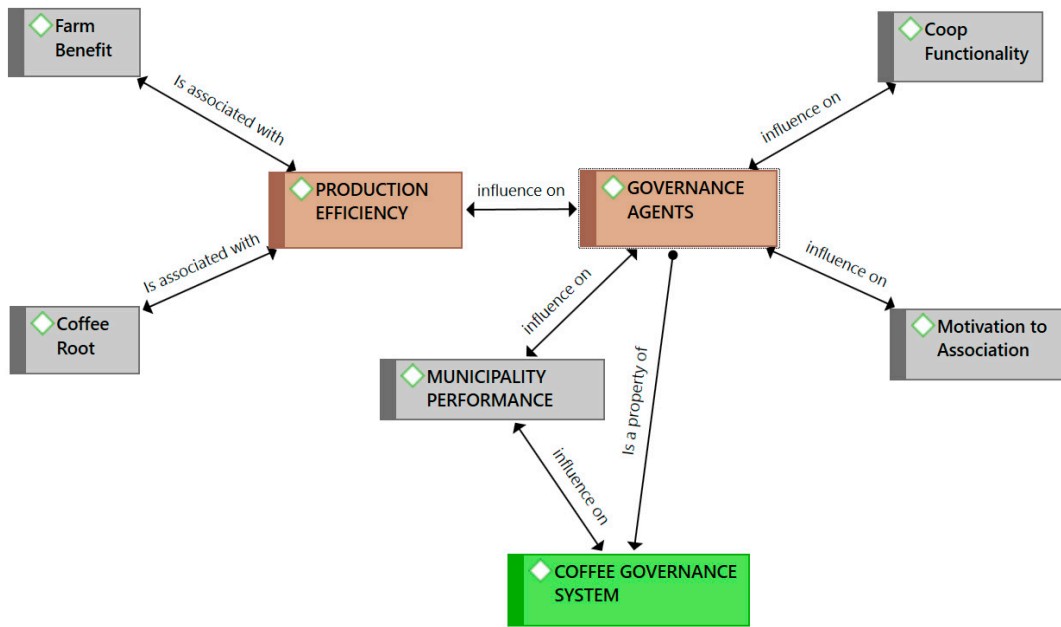

**Figure 5.** Tangible and intangible components of the TG associated with coffee. The author.

In Sections 3.1 and 3.2, the findings of the quantitative part are discussed, in Section 3.3 the qualitative findings are presented. Finally, Section 3.4 is dedicated to merge both analyses and provide the implications of the study.

## 4. Results and Discussion

### 4.1. Exploratory Factor Analysis (EFA)

As the main result of this stage, it is reported that 15 out of 16 variables met the assumptions of normality of the variance. The factor loadings and principal components resulting from having executed the EFA procedure are presented in Tables 2 and 3. EFA results are critical to moving forward with subsequent SEM analyses.

**Table 3.** Definition of components and percentage of explained variance.

| Components (Constructs) | Total Variance Sums of Charges Squared by Rotation | | |
| --- | --- | --- | --- |
| | **Total** | **% Variety** | **% Accumulated** |
| 1. Governance Agents | 4.591 | 28.69 | 28.69 |
| 2. Production Efficiency | 4.185 | 26.16 | 54.85 |
| 3. Municipality performance | 2.80 | 17.51 | 72.36 |

Extraction method: principal component analysis.

Table 3 lists the components resulting from the Varimax extraction method. Each component shows the percentage of variance that can be explained.

Tables 2 and 3 provide some information about collinear variables and relationships with their respective components (latent variables). Component one is loaded into 10 out of 15 variables, of which we only consider those seven variables that obtained a factor load greater than 0.5 since this shows a high correlation. This component one can explain 28.69% of the total variance, which denotes a strong association with the construct (unobserved variable): GA. Component two is sufficiently loaded into five variables that explain 26.16% of the variance and show its close association with the construct: PE. Four variables define component three that explains 17.51% of the variance and its association with the municipality performance. In this group of correlations, only the housing deficit variable charges

negatively with a value of −0.856, which explains that as this deficit increases, welfare conditions will be harmed. The three components add up to 72.36% of the variance that can be explained from the data panel subjected to this EFA analysis. Several authors agree that a factor load greater than 0.5 proves the existence of a sufficient correlation or dependency between the observable variables and their respective latent variables [68,70].

In other words, the GAs of component one would increase their territorial presence in direct proportion to the observable variables of productivity, associativity, and cooperativity. However, PE efficiency would be improved with the increase in organization, participation, and coffee vocation. Variables of coverage of public services are directly associated with the better performance of the municipality.

Additionally, if the variables that make up each component in Table 2 are compared with the original grouping of variables in Table 1, it can be noted that once the EFA was executed, certain changes occurred in the arrangement of observable and latent variables with respect to what was proposed from the initial theoretical design. For example, the organizational and participation variables were believed to be more associated with GA. However, the EFA determined that there is a higher factorial linkage with production efficiency. Similarly, road investment is more associated with the municipality's performance than GA. This re-arrangement is a normal adjustment when the EFA analysis is executed, and changes can only be accepted to the extent that they have a theoretical explanation or due to the observed reality [72,73]. In this case, it is justified by the fact that the coffee boards are oriented towards the production efficiency of their associated producers. Something that shows to be dependent on the capacity of organization and participation of the producers [61,66].

## 4.2. Confirmatory Factor Analysis (CFA)

This is the stage where SEM analysis plays a fundamental role. To meet the reliability and validity criteria of the coffee governance model under study, it is essential that the dataset entered complies with the quality of fit measures. The CFA allows for consolidation of the work previously done in EFA leading to a greater contrast of the specified hypotheses [45,68]. Through AMOS, the "path diagram" analysis was carried out to verify if this pattern corresponds to the observed reality and modeled by the theoretical design studied [73].

Figure 3 shows the model fitted in a specific pattern that focuses on the three latent variables analyzed (in ovals). Values greater than 0.5 indicate a strong link with their respective independent variables (in boxes). PE, for example, shows a strong dependence of the variables: Org. with 0.77, Part with 0.90, Vprod with 0.96, and Aprod with 0.83 according to this model. The same can be said about the other latent variables examined.

The individual reliability test is the relationship between each independent variable (rectangle) and its respective dependent variable (oval). It can be seen in Figure 3 that all the standardized loads of the multiple regression exceed values of 0.5 as proof of dependence. Loads less than 0.5 indicate that the general model will be adversely affected [68,73]. Table 4 shows the results of six tests of convergent validity applied by AMOS. The obtained values were subjected to the SEM model test analysis. Completion of this phase allows us to determine if the constructs represented in the coffee governance model of the territory under study are correctly measured by the indicators [45,69], see Figure 3.

**Table 4.** Results SEM/AMOS tests for the coffee governance model.

| Adjustment Index | Initials | Expected | Results |
|---|---|---|---|
| Chi$^2$ | | >0.05 | 122.520 |
| Discrepancy between Chi$^2$ and degrees of freedom | CMIN/DF | ≤5 | 2.112 |
| Comparative Adjustment Index | CFI | 0.90–1 | 0.935 |
| Non-Standard Adjustment Index | TLI | 0.90–1 | 0.913 |
| Normalized Adjustment Index | NFI | 0.90–1 | 0.887 |
| Root mean square error of approximation | RMSEA | ≤0.08 | 0.111 |

At this point, it is worth remembering a great virtue of the SEM/AMOS analysis in terms of revealing, by means of the variance error, the measure of the underlying variables (ovals). These variables, according to statistical theory, would be endogenous if in the model they appear depending on other variables, such as the case of PE, MP, and GAs. The variable coffee governance system (lower oval) would be exogenous since the effect of endogenous variables is directed towards it [45], thus explaining how this endogenous model contributes to determining this governance system associated with the coffee territory. The values that appear between the endogenous and exogenous variables represent the error of the variance. It should be interpreted similarly to the respective coefficient calculated in a correlation, being a measure of how much they influence each other. This value also has a scale from 0 to 1.

In this sense, Figure 3 also shows that the weight or degree of influence exerted by the GA to contribute to PE is 61%, while the MP in terms of public works is influenced by 19%. Similarly, it can be seen that the exogenous variable is doubly favored both by the GA with an intensity (error of the variance) of 31% and by the MP with an intensity of 97%.

The reliability tests of variables are examined with their respective "path diagram" in Figure 3 presented in Table 4. They indicate that four out of six exams were passed. This provides evidence that the four-component model represents, with acceptable precision, the reality studied [70]. Statistically, it defines how the coffee governance system works, at least in western Honduras.

If these data are contrasted with the available theoretical evidence on the role played by the GA, it is understandable because they affect PE. These associative networks deploy services that improve market conditions, price management, financing, exchange of technical information, and more [5,45]. The same logic is observed regarding the MP. This performance improves when the production part is improved. This increase favors the economy and more public works can be financed [4].

*4.3. Work with ATLAS.ti*

From the qualitative part, it will be seen that the metric of the SEM analysis has agreement with the field inquiries, as shown in Figure 4.

The main contribution from this qualitative perspective was the identification of those invisible forces associated with the governance and that mobilize the GAs, such as the motivation to associate and their roots in the coffee territory. This broadens the explanation of why the producer, despite difficult times, clings to the coffee plantation as a lifestyle. In interviews, they use expressions like: "it's my way of living" and "every day it's worth returning to ours" that reveal their passion and territorial identity. Similarly, expressions such as "associates we are stronger" and "we organize ourselves so that large coffee merchants do not take advantage", "as cooperative members, they respect us more" define the motivation of many producers towards cooperative membership. This is important because it reaffirms that the governance of the coffee growing space, both in the theorized design and in the experience obtained through interviews, is mainly represented by networks of producers affiliated and organized in different cooperatives and rural boards. In both perspectives, it is understood that things like identity (roots), motivation, expectations of benefits of the farm, and the functionality of the cooperatives would make the governance agents manifest themselves. The motivation behind this would be the interest in generating improvements in productive and commercial efficiency, access to

better markets [50,54], also improvements to the production environment such as good roads, electricity, and water coverage; health and education, especially for the family nucleus of the primary producer and the local labor force [48].

This interaction between actors necessarily extends to the social and institutional sphere of government because producers are also citizens, engendering an indissoluble relationship between production and community [17,64]. This reinforces the network of actors, who, thanks to the testimony of goals achieved, increase trust and satisfaction with associativity [36,41].

Additionally, the concepts that explain the identity or coffee roots in the territory are reinforced with the expectations of the farm's benefits. All of this tends to influence PE. On the other hand, the evidence of cooperative functionality in terms of access to new and better markets and prices motivates them to join with a growing membership, resulting in stronger organizations [46,67]. At the territorial level, these capacities and qualities will influence better well-being conditions in the communities. Therefore, it can be said that the statements proposed in the study's hypotheses are significantly supported by the findings of this living thought referred to in grounded theory [78].

Figure 5 shows the dimensions or variables studied both in SEM and in the qualitative part. This shows interrelation with the intangible elements revealed by the interviewees. It also gives way to an enriched scheme that in no way contradicts what was calculated in the quantitative part. This broadens the understanding of what makes the coffee governance system work.

### 4.4. Implications

The findings provided by this methodological combination have substantially penetrated the nature of the TG phenomenon, not only in capturing its direct and indirect effects, as the majority of the literature has reported; but this study has also revealed new intangible elements (rooted in the territory, motivation, and expectations towards associativity) that gravitate to the protagonist territorial actors. Something that helps to explain why and how this complex process of human interaction is being built, at least in the coffee-growing and rural territory addressed.

The main implication of this study as a tool for the analysis of Territorial Governance Systems is its ability to better inform political decisionmakers who manage rural development in developing countries. This analysis can be replicated both in the other coffee-growing territories of developing countries, as well as in those rural environments that have defined their own livelihoods. This approach shows the qualities acquired by those territories that are more informed and cohesive to prevent the mechanisms that generate poverty in rural areas, directing in a more correct way towards sustainable and long-term community development.

## 5. Conclusions

The statistical model proposed from the SEM exam describes a specific pattern of territorial dynamics that are favored by different coffee organizations, conceptualized here as governance agents. By contrasting the two hypotheses of the study with the results obtained, we see that both have been verified. Thanks to the governance agents, production efficiency is positively affected (H1). This is true to the extent that the variables associated with coffee cooperatives and boards' increase in membership and are consolidated in producer participation. This generates a favorable effect on production efficiency in terms of yield, which, in the case of the study area, is estimated at a proportion of 0.61 according to the measurement of the variance obtained. Regarding (H2), the endowments that a municipality achieves show improvement in a proportion of 0.19, depending on the productive improvement that is being acquired. More important still would be the conformation of a governance system (exogenous variable) whose implication favors rural coffee development. This would create benefits in two ways, given that the variance dimensions of both the performance of the municipality and the governance agents reveal a positive endogenous effect.

If we triangulate the findings of both parts of the analysis, it can be seen from the quantitative part that the statistical relationships between the four components of the governance model show a recursive

interaction that is mediated by positive correlations. While in the qualitative part, those triggers that facilitate associativity and cooperativity are revealed, determining as a whole how this system works in western Honduras. In turn, the recursive aspect would explain a kind of virtuous circle, which is internally reinforced as each of the dimensions of the model is consolidated. This would undoubtedly have a favorable impact on rural development, provided that the modeling of a governance system becomes a development strategy.

The challenges facing the rural world are a concern perhaps without a satisfactory response. This research confirms the interest of governance systems as an agglutinating strategy for territorial resources. This strategy could be an effective means of reaching the rural poor, as well as being an ally for state programs and external cooperation aimed at rural well-being. The strategy seems to be viable in the light of the theoretical and empirical findings presented, supporting the affirmation that sustainable rural development should consider human qualities and their territorial worldview as key elements of any solution formula.

Finally, the argumentative and methodological structure of this article is not without limitations. The analysis of the agricultural linkage that exceeds the territorial actors and the balance of the bargaining power that occurs between them were not analyzed in this study and are proposed as avenues for future research.

**Author Contributions:** Conceptualization, O.M.P.; methodology, O.M.P., J.M.D.-P.; formal analysis, O.M.P., J.M.D.-P., J.L.Y.; writing—original draft preparation, O.M.P., J.M.D.-P., J.L.Y.; writing—review and editing, O.M.P., J.M.D.-P.; supervision, J.M.D.-P. All authors have read and agreed to the published version of the manuscript.

**Funding:** This research received no external funding.

**Conflicts of Interest:** The authors declare no conflict of interest.

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
