# Peer review of "The Role of Coffee Organizations as Agents of Rural Governance: Evidence from Western Honduras"

_land, doi:10.3390/land9110431_

Round 1

Reviewer 1 Report

Figure 1: Governance agents is(are) a property of the system. So, they belong to the system. Fine. However, this is not clear from lines around the figure to the reader. The design/relations of Figure 1 should be explained more.

Reviewer 2 Report

Congratulations for a very well-presented article. You do a very good job in explaining theoretical base, hypotheses, research design and methods applied.

However, I believe the qualitative part is a bit neglected in comparison to the quantitative analysis. I appreciate the idea of triangulating methods, but you surely could use your wealth of data (76 interviews!) to support your argument and provide both more evidence and context to your findings.

For context, it would also be great to get more information on the nature of value chains and the intricacies of Honduran coffee as a commodity, also a sentence or two on the history of coffee cultivation and associated rural settings would be helpful.

Sometimes I find your writing style a bit too colorful, e.g. ln 192-194 "It is an army of producers who carry out this task immersed in the folkloric and social aspects of the rural landscape"

This sentence does not make sense: "who are prone to join cooperatives and other forms organizational coffee" (ln. 158/159)

Finally, in your conclusion, you do not discuss wether the findings can be possibly applied to coffee cultivation in other areas, and also, I find that parts of the conclusion are too general to be meaningful (ln 433-439).

Reviewer 3 Report

The Role of Coffee Organizations as Agents of Rural Governance: Evidence from Western Honduras

The following comments/suggestions should be addressed to improve the quality of the paper

Abstract

  • The last paragraph is incomplete because it lacks a verb.
  • It should end by mentioning the implication or conclusion of the study findings.

Introduction

  • Line 28: use “developing countries,” which is better than “underdeveloped countries”
  • Highlighting that about 2.6 billion people directly rely on agriculture for livelihoods (Sustainable Development Goal [SDG] report 2018) will help substantiate the first sentence.
  • Line 28-31: What prevailing “economic model” and what are the examples of the “market barriers?”
  • Line 36: define the acronym TG here despite doing so in the abstract.
  • Line 44: this is redundant: public government
  • Line 83: Again, define SEM here
  • Mention the knowledge gap the paper is trying to fill and its contribution to the literature
  • The last two paragraphs are about methods. They are better placed in the Materials and Methods section

Theoretical background

  • Better change this to section 2, divided into: 2.1. Agents of rural governance; and 2.2. The systemic character of governance.
  • Underline that, in 2015, about 783 million people in the world were living below USD1.90 a day (World Bank). Also, the SDG 1 targets ending poverty in all its forms by 2030
  • Please enlarge the font size of texts in Figure 1-2 and Figure 5 to make them more legible.
  • Line 152: use “conceptual perspective” rather than “theoretical perspective”
  • Line 156: replace “design” with “framework”
  • Lines 185-200: section 2.1 should be merged with the Materials and Methods section to form section 3.1

Materials and Methods

  • Section 2.1: Justify selecting 16 indicators and 4 constructs.
  • Line 65: What makes the interview “random”? the use of a random table or probability table or what?
  • Lines 66-67: Are the 76 surveys the same as in-depth interviews?
  • Section 2.1: State how you recruited the interview participants as well as the inclusion/exclusion criteria used
  • Section 2.1: Who conducted the interviews, in which language, when (the month and year), and where (settings)?
  • Section 2.1: How were the interviews recorded and ethical issues addressed (e.g. consent and confidentiality)?
  • How was the grounded theory/analysis used in analyzing the interview transcript?
  • I recommend reviewing the data collection and analysis section of the following papers for a precise example of how in-depth interviews and grounded analysis are reported: https://doi.org/10.1080/02508060.2018.1490862
    https://doi.org/10.1177/1558689817710877
    https://doi.org/10.1016/j.surg.2020.04.056

Results

  • Line 282: “analyses” rather than “analyzes”
  • Line 354: enlarge Figure 3
  • Lines 405-409: Implications of a study are placed in the discussion or conclusion section. Replace this by summarizing the new/important findings?

Discussion and conclusion

  • Line 410: correct the spelling of the word conclusion.
  • There is not a single citation in the discussion section, which should show how the findings corroborate or differ from prior studies and the likely explanations.
  • The discussion should relate the findings to the research hypothesis raised earlier.
  • It should underscore the value the study added to the literature.
  • To what extent can the method be applied to other settings?
  • Are there analytical and/or theoretical implications of the findings?
  • What are the implications of the findings for rural governance policy?
  • What are the limitations of the study and future research direction?

Round 2

Reviewer 3 Report

The following few methodological issues are yet to be resolved:

  • Lines 305-309: How were the 76 coffee producers selected (sampling technique)?
  • What are the locations of the interviews? On their farms or where?
  • Was an interview schedule or protocol used?
  • How were the interviews recorded: using an audio recorder or by paper and pencil?
  • Atlas Ti is just a tool not a technique of qualitative data analysis. Content analysis, grounded theory/analysis are examples of such techniques. 
  • For example, the grounded analysis consists of open coding, axial coding, and selective coding.
  • I recommended some papers that can help to improve the methodology section, but they are ignored.
  • Lines 320-321: there is a repetition here "All this information was coded in 7 categories (see figure five), all this information was coded into 7 categories"
  • Check and fix other redundancies to make the paper more succinct.
